# Inhibition of the Heat Shock Protein A (HSPA) Family Potentiates the Anticancer Effects of Manumycin A

**DOI:** 10.3390/cells10061418

**Published:** 2021-06-07

**Authors:** Damian Robert Sojka, Sylwia Hasterok, Natalia Vydra, Agnieszka Toma-Jonik, Anna Wieczorek, Agnieszka Gogler-Pigłowska, Dorota Scieglinska

**Affiliations:** 1Center for Translational Research and Molecular Biology of Cancer, Maria Sklodowska-Curie National Research Institute of Oncology Gliwice Branch, 44-102 Gliwice, Poland; damian.sojka@io.gliwice.pl (D.R.S.); hasterok.sylwia@gmail.com (S.H.); natalia.vydra@io.gliwice.pl (N.V.); agnieszka.toma-jonik@io.gliwice.pl (A.T.-J.); agnieszka.gogler-piglowska@io.gliwice.pl (A.G.-P.); 2Division of Medical Biology, Institute of Biology, Jan Kochanowski University, 25-406 Kielce, Poland; anna.wieczorek@ujk.edu.pl

**Keywords:** heat shock factor 1, heat shock proteins, HSP70, HSPA1, HSPA2, manumycin A, lung cancer, breast cancer, HSPA inhibitors

## Abstract

Manumycin A (MA) is a well-tolerated natural antibiotic showing pleiotropic anticancer effects in various preclinical in vitro and in vivo models. Anticancer drugs may themselves act as stressors to induce the cellular adaptive mechanism that can minimize their cytotoxicity. Heat shock proteins (HSPs) as cytoprotective factors can counteract the deleterious effects of various stressful stimuli. In this study, we examined whether the anticancer effects of MA can be counteracted by the mechanism related to HSPs belonging to the HSPA (HSP70) family. We found that MA caused cell type-specific alterations in the levels of HSPAs. These changes included concomitant upregulation of the stress-inducible (HSPA1 and HSPA6) and downregulation of the non-stress-inducible (HSPA2) paralogs. However, neither HSPA1 nor HSPA2 were necessary to provide protection against MA in lung cancer cells. Conversely, the simultaneous repression of several HSPA paralogs using pan-HSPA inhibitors (VER-155008 or JG-98) sensitized cancer cells to MA. We also observed that genetic ablation of the heat shock factor 1 (HSF1) transcription factor, a main transactivator of HSPAs expression, sensitized MCF7 cells to MA treatment. Our study reveals that inhibition of HSF1-mediated heat shock response (HSR) can improve the anticancer effect of MA. These observations suggest that targeting the HSR- or HSPA-mediated adaptive mechanisms may be a promising strategy for further preclinical developments.

## 1. Introduction

Investigations of natural products as a source of bioactive compounds are becoming an important research area in the discovery and development of drugs. There are numerous products derived from plants or microorganisms that have exhibited anti-carcinogenic activities by interfering with the initiation, development, and progression of cancer. Marine microorganisms are a rich source of novel pharmacologically active metabolites, among which actinomycetes hold a prominent position. It is noteworthy that actinomycetes produce about 70% of the world’s naturally occurring antibiotics, exhibiting either antimicrobial (antibacterial, antifungal, antiprotozoal), antitumor, and/or antiviral activities [1,2]. Manumycin A (MA), a natural antibiotic (secondary metabolite) belonging to a class of polyketide compounds, was isolated from a marine *Streptomyces* sp. M045, derived from sediment of Jiaozhou Bay in China. To date, MA has shown anticancer activity in various experimental in vitro models of fibrosarcoma [3], pancreatic cancer [4], anaplastic thyroid cancer [5], triple-negative breast carcinoma [6], colorectal cancer [7], and others. Interestingly, MA exhibited an anticancer effect both as a single agent and in combination with various anticancer drugs in several preclinical in vivo cancer models. Initially, the anticancer mechanisms of MA were attributed to its ability to inhibit RAS-mediated signaling via blocking of G-protein farnesylation [3,8]. However, MA exerts pleiotropic anticancer effects by targeting multiple proteins and signaling pathways. MA directly influences the expression and activity of several important transcription factors (e.g., NF-κB, Sp1, p53) [9,10,11]. MA can stimulate reactive oxygen species generation through mechanisms that rely on increased superoxide anion production [12,13]. Recently, MA has also been reported as an inhibitor of exosomes biogenesis and release ([14], reviewed in [15]).

In order to minimize toxic effects of anticancer treatment, cancer cells are capable of triggering the adaptive stress responses. One of the well-known cytoprotective mechanisms is the so-called heat shock response (HSR), which is orchestrated by the selective activation of heat shock transcription factor 1 (HSF1). HSF1 stimulates transcription of genes that encode chaperones, primarily from the heat shock proteins (HSPs) families. The general function of HSPs is to assist with protein folding either during de novo synthesis or under stress conditions, thus preventing excessive protein denaturation and aggregation as well as protecting against apoptosis [16,17]. The human HSPs, according to their molecular weight and function, are divided into several families: HSPH (HSP110), HSPC (HSP90), HSPA (HSP70), DNAJ (HSP40), and HSPB (small HSP) [18]. The human genome contains thirteen highly homologous *HSPA* genes. The classical representatives are *HSPA1A* and *HSPA1B* genes, both coding for the HSPA1 protein, the best studied stress-inducible and cytoprotective member of the HSPA family. Promoters of these genes contain a heat shock element (HSE), a cis-regulatory sequence that is indispensable for HSF1 binding and stress-inducible gene expression. However, the expression of several other *HSPA* genes, due to a lack of the HSE sequence, cannot be elevated in a HSF1-dependent manner. An exemplary representative is *HSPA2*, the gene coding for a testis-enriched chaperone with a critical role for spermatogenesis [19] but also implicated in modulating the phenotype and differentiation of epithelial cells [20,21]. The knowledge about the contribution of HSPA2 to the cytoprotective system of the cells has not been fully elucidated.

HSPA proteins are overproduced in various types of cancer cells lines and tumors. They are regarded as potent cancer-related pro-survival factors that can counteract the deleterious effects of various stressful stimuli (including anticancer drugs) or environmental conditions [22]. It is well established that cancer-related HSPA chaperones serve to support growth and/or inhibit cell death pathways. Therefore, HSPAs have recently emerged as one of the potential therapeutic targets [23]. Previously, it was speculated that the induction of HSPAs (HSP70) expression may provide cytoprotection against MA-induced toxicity [24]. However, to date, no direct confirmatory evidence has been provided. It is worth noting that the HSPs induction may have limited (if any) survival benefit in cells exposed to chemotherapeutics. For example, the lack of an evident relationship between HSPs levels/induction and cells sensitivity to platinum derivatives was reported (reviewed in reference [25]). Therefore, this study was undertaken to examine whether targeting HSPAs- or HSF1-mediated cytoprotection would increase the effectiveness of MA in lung and breast cancer cells.

## 2. Materials and Methods

### 2.1. Cell Culture and Experimental Conditions

Lung cancer cell lines NCI-H1299 (non-small cell lung carcinoma, CRL-5803), NCI-H23 (adenocarcinoma, CRL-5800), A549 (lung carcinoma, CCL-185), and MCF7 (breast adenocarcinoma, HTB-22) were purchased from ATCC (Manassas, VA, USA). Cells were cultured at 37 °C under standard conditions (5% CO_2_, 95% humidity, 21% O_2_ concentration) in RPMI (NCI-H1299, NCI-H23; Sigma-Aldrich, Merck KGaA, Darmstadt, Germany), DMEM-F12 (MCF7; Sigma-Aldrich, Merck KGaA, Darmstadt, Germany) or DMEM HG (A549; Sigma-Aldrich, Merck KGaA, Darmstadt, Germany) medium supplemented with 10% fetal bovine serum (EuRx, Gdańsk, Poland) and antibiotics (gentamycin or penicillin-streptomycin). All experiments were performed in the absence of antibiotics. Cells were regularly checked for mycoplasm contamination.

### 2.2. Incubation Experiments

The following stock solutions were used: Manumycin A (MA) (10 mM in DMSO; Sigma-Aldrich, Merck KGaA, Darmstadt, Germany), VER-155008 (VER) (20 mM in DMSO; Sigma-Aldrich, Merck KGaA, Darmstadt, Germany), JG-98 (1.5 mM in DMSO; a kind gift of Prof. M.Y. Sherman (Ariel University, Ariel, Israel). Working solutions were prepared fresh before each experiment in culture medium (without antibiotics). Cells were incubated with 0–50 µM MA, 0–15 µM VER, and 0–1.5 µM JG-98. Control cells were incubated with medium containing DMSO.

### 2.3. Cell Viability and Cytotoxicity Assays

The effect of drug treatment on cell viability was determined using MTS or alamarBlue^®^ assays. MTS assay was performed using CellTiter 96 Aqueous One Solution reagents according to the manufacturer’s protocol (Promega; Madison, WI, USA). Briefly, cells (3 × 10^3^, NCI-H1299; 5 × 10^3^, NCI-H23; 2.5 × 10^3^, A549; 4 × 10^3^, MCF7 cells per well) were plated into 96-well plates and incubated with chemical regents for 72 h. Absorbance of the formazan product was measured (λ = 490 nm) using a microplate reader. Absolute IC_50_ values (mean ± 95% confidence intervals (CI)) for MA were calculated by fitting the dose-response curve using GraphPad Prism Software (GraphPad Software; La Jolla, CA, USA). For the alamarBlue^®^ assay, cells were seeded into 96-well plates (4 × 10^3^ cells per well) and incubated with manumycin A for 72 h. Following the incubation period, alamarBlue^®^ solution was added according to a manufacturer’s protocol (Invitrogen; Life Technologies; Carlsbad, CA, USA), and after 1 h the fluorescence intensity of alamarBlue^®^ reagent at Ex/Em 570/600 nm was measured using a microplate reader. The cytotoxic effect of MA alone or in combination with VER were evaluated using the CellTox™ Green Cytotoxicity Assay (Promega; Madison, WI, USA). MCF7 cells were seeded into 96-well plates (4 × 10^3^ cells per well) and exposed to appropriate compound(s) in the presence of CellTox™ Green Dye for 72 h according to the manufacturer’s protocol. Following the incubation period, the fluorescence intensity of the CellTox^TM^-stained cells at Ex/Em 485/528 nm was measured using a microplate reader.

### 2.4. Protein Extraction and Western Blot Analysis

Cells were seeded in 6 cm dishes with a maximum confluence of 50–60%, and 24 h after plating, they were exposed to the drugs or inhibitor for 24 h. To prepare total protein extracts, cells were lysed by scrapping in RIPA buffer (1× PBS, 1% NP-40, 0.1% SDS, 0.5% SDC, 50 mM NaF, 1 mM PMSF) supplemented with a protease inhibitor mixture (Roche Molecular Systems, Inc; Rotkreuz, Switzerland). After incubation on ice (15 min), lysates were centrifuged (4 °C for 15 min at 22,000× *g*). Total protein content was determined using a Protein Assay Kit (Bio-Rad; Hercules, CA, USA). Then, 25–35 μg of total proteins were fractionated by SDS-PAGE on 8–10% polyacrylamide gels and transferred onto nitrocellulose membrane using a Trans Blot Turbo system (Bio-Rad; Hercules, CA, USA) for 10 min. The membrane was blocked (60 min) in 5% nonfat milk/TTBS (0.25 M Tris–HCl (pH 7.5), 0.15 M NaCl, and 0.1% Tween-20) and incubated (overnight at 4 °C or 1 h at 37 °C) with primary antibodies (Appendix A). The antibody–antigen interaction was detected using secondary antibodies and visualized using Clarity ECL Western Blot Substrate (Bio-Rad; Hercules, CA, USA). Immunodetection of β-actin was used as a protein loading control.

### 2.5. Generation of Genetically Modified Cell Lines

The construction of the lentiviral vectors coding for the control non-targeting shRNA and lentiviral shRNA targeting the coding sequence of human *HSPA2* (Entrez Gene: 3306) and *HSPA1A*/*HSPA1B* (Entrez Gene: 3303/4) genes were described in detail previously [26]. Briefly, double-stranded oligonucleotides (shRNA sequences are available in [26]) were inserted into the pLVX-shRNA1 vector (Clontech/Takara Bio, Mountain View, CA, USA). The construction of a lentiviral vector encoding the HSPA2 protein under the CMV promoter in the pLVX-Puro plasmid (Clontech/Takara Bio, Mountain View, CA, USA) was described previously [27]. Generation of infectious lentiviruses and cells transduction were performed according to the manufacturer’s instructions (Clontech/Takara Bio, Lenti-X shRNA Expression System, Mountain View, CA, USA). Stably transduced cells were enriched by puromycin selection. The control cell lines were generated by subsequent transductions of cells with pLVX-shRNA1 vectors encoding shRNA-luc sequence or pLVX-Puro vector.

### 2.6. HSF1 Functional Knockout Using the CRISPR/Cas9 Editing System

Edit-R Human HSF1 (3297) crRNA, Edit-R tracrRNA, and Edit-R hCMV-PuroR-Cas9 Expression Plasmid (Dharmacon, Lafayette, CO, USA) were introduced into MCF7 cells using DharmaFECT Duo (6 µg/mL) (Dharmacon, Lafayette, CO, USA) according to the producer’s instruction. Transfected cells were enriched by puromycin selection for 4 days. Afterwards, single clones were obtained by limiting dilution on a 96-well plate. Out of 81 clones, two individual HSF1-knockout clones (CRISPR-K14 and CRISPR-K45) and six pooled control clones (CRISPR-CTL) were chosen for the subsequent experiments. Two HSF1-targeting crRNAs (crRNA GTGGTCCACATCGAGCAGGG; crRNA AAAGTGGTCCACATCGAGCA; both in exon 3 on the plus strand) out of the five tested led to the efficient knockout of the *HSF1* gene expression.

### 2.7. Statistical Analysis

Unless stated otherwise, all data are shown as mean ± standard deviation (SD) of the mean. Difference significance between two groups was determined by two-tailed *t*-test for independent samples. The *p*-value of less than 0.05 was considered statistically significant. Excel 2013 (MS Office, Microsoft, Albuquerque, NM, USA) was used for the analyses.

## 3. Results

### 3.1. Manumycin A Inhibits the Growth of NSCLC Cells in a Cell Line-Dependent Manner

We compared the viability of three human lung cancer cell lines, namely NCI-H23, NCI-H1299, and A549, following exposure to increasing doses of MA. These cells were characterized by different levels of approved MA molecular targets. A549 cells, as compared to NCI-H1299 and NCI-H23 cells, expressed significantly lower levels of FTα and FTβ farnesyl transferases and Sp1 transcription factor (Figure 1a). Total levels of RAS (pan-RAS) were lower in NCI-H1299 and A549 cells than in NCI-H23 cells (Figure 1a). All cell lines carried RAS mutations (KRAS, A549 and NCI-H23 cells; NRAS, NCI-H1299 cells).

Results of the MTS assay revealed that the cells exhibited differential cell line-specific sensitivity to MA (Figure 1b). The IC_50_ value of MA on A549 cells (21.65 µM; 95% CI, 19.98–23.17) was higher than that on NCI-H23 (8.71 µM; 95% CI, 8.36–9.09) and NCI-H1299 cells (9.0 µM; 95% CI, 8.68–9.51) (Figure 1b).

### 3.2. MA Modulates HSPAs Expression in Lung Cancer Cells

To search for the molecular mechanisms responsible for the different responses to MA, we examined the MA-induced changes in expression of HSPA proteins, which are well-known indicators of HSR activation. We found that MA led to a dose-dependent upregulation of the stress-inducible HSPA1 and HSPA6 paralogs. HSPA1 is a major stress-inducible paralog, which can be produced at some basal level in non-stressed cells, while HSPA6 represents a strictly stress-inducible paralog (Figure 2a and Appendix A). However, MA-sensitive cells (NCI-H1299 and NCI-H23) showed higher magnitude of HSPAs upregulation by MA at doses below IC_50_ value in relation to that of A549 cells. In the latter cell line, only modest induction was detectable by MA at doses much above the IC_50_ value.

MA treatment also resulted in alterations in the protein levels of non-stress-inducible HSPAs (Figure 2a). First of all, all cell lines showed a reduction in HSPA2, the poorly characterized paralog expressed in a cell-type specific manner (Figure 2a). Other cell type-specific changes included a reduction in HSPA9 (mortalin, GRP75), a paralog primarily localized in the mitochondria in NCI-H23 and NCI-H1299; a reduction in the protein level of HSPA8, a constitutively expressed cytoplasmic/nuclear paralog in NCI-H23 cells; and upregulation of HSPA5, the endoplasmic reticulum-localized paralog in NCI-H1299 cells (Figure 2a, Appendix A). Altogether, we found that MA caused a concerted pattern of changes in the protein level of HSPAs, an upregulation of the inducible HSPA1 and HSPA6, and downregulation of HSPA2 and HSPA9 encoded by non-stress-inducible genes in MA-sensitive NSCLC cells.

Because expression of the inducible HSPAs is regulated by HSF1, we studied the phosphorylation of HSF1 at Ser326, which is crucial for transcriptional competence of this transcription factor under stressful conditions [28]. We found prolonged phosphorylation of HSF1 at serine 326 lasting for up to 9 h in cells exposed to toxic doses of MA (10 µM for NCI-H1299; 30 µM for A549 cells) (Figure 2b). In MA-treated cells, total levels of HSF1 remained stable (Figure 2b). In A549 cells, a non-toxic concentration of MA (10 µM) resulted in transient phosphorylation of HSF1 (Figure 2b). Altogether, these results indicate that treatment with MA led to HSF1 activation and subsequent induction of the HSR in NCI-H23 and NCI-H1299 cells. MA-induced activation of HSF1 was confirmed by increased transcription of the HSF1-dependent *HSPA1A* and *HSPA6* genes as revealed by RT-qPCR (Appendix A). This MA-induced response seems to be less effectively activated in A549 cells.

### 3.3. HSPA1 Exerts Very Limited, Cell Type-Dependent Protection against MA Toxicity While the Role of HSPA2 in Cytoprotection Is Negligible

We found that MA-treated cells showed two-directional changes in the protein level of HSPA paralogs (Figure 2a). Thus, we examined whether these responses are cytoprotective in MA-treated cells. To discover the potential interdependence between HSPA1 (upregulated under MA treatment) and HSPA2 (downregulated under MA treatment) in regulating cell susceptibility to MA, we examined effects of downregulation of HSPA1 expression or upregulation of HSPA2 in MA-sensitive cells. We selected these paralogs since they are regarded as important cancer-related chaperones; HSPA1 is believed to have potent cytoprotective roles, while contribution of HSPA2 to cytoprotection is debatable. We used modified NCI-H1299 cell lines with knockdown of *HSPA1A*/*B* genes expression (sh-A1.N and sh-A1.S) or with HSPA2 overproduction (p-A2) (Figure 3a,b). The other model was NCI-H23 cells with either knockdown of *HSPA1A/B* expression (sh-A1.N and sh-A1.S cell lines; Figure 3c,d) or of *HSPA2* expression via the shRNA-mediated mechanism (sh-A2.3 and sh-A2.4 cell lines; Figure 3c,d). We decided to downregulate HSPA2 levels in NCI-H23 cells because they contain higher endogenous levels of HSPA2 in relation to that of NCI-H1299. The cell lines produced comparable levels of HSPA1 (Appendix A). Both models were described in detail in our previous papers [21,26,27], where we found modulation of HSPA1 or HSPA2 expression levels had no effect on proliferation of NSCLC cells [21,26].

We found that NCI-H1299 cells deficient in HSPA1, but not NCI-H23 cells, were more sensitive than were control cells to a low dose (5 µM) but not to higher doses (7.5–10 µM) of MA (Figure 3b). HSPA2 had negligible effects on cells sensitivity to MA since neither its high overproduction (Figure 3a,b) nor significant downregulation (Figure 3c,d) affected the viability of MA-treated cells. Altogether, our results indicated that HSPA1 alone may provide some protection against MA, but this effect is very limited and cell type dependent.

### 3.4. Inhibition of HSPAs Sensitize Lung Cancer Cells to MA

Given that selective knockdown of major stress-inducible *HSPA1* expression evoked a very limited MA-sensitizing response, we decided to examine whether simultaneous inhibition of several HSPA paralogs would potentiate the antiproliferative effect of MA on lung cancer cells. We used VER and JG-98, two pan-HSPA small molecule inhibitors that bind non-selectively to several HSPA paralogs but show different mechanisms of action [29,30]. Antiproliferative effects of VER and JG-98 on lung cancer cell lines were reported by us earlier [26]. In this study we combined MA with pan-HSPA inhibitors; all chemical compounds were used at concentrations corresponding to IC_50_ values or lower. As expected, each pan-HSPA inhibitor reduced metabolic activity of cells in a dose-dependent manner (Figure 4). The concurrent action of MA and the pan-HSPA inhibitor evoked significantly greater antiproliferative effects than those with each drug separately (Figure 4a–e). We examined cytotoxicity of single and combined drug treatments using the CellTox™ cytotoxicity assay (detection of cell membrane permeability). Results in Figure 4f,g are in agreement with our previous study [26] showing that VER activated cell death in NCI-H23, but not in NCI-H1299 cells. Single treatment with MA (at doses below IC_50_ value) was cytotoxic in NCI-H23 cells, but not in NCI-H1299 cells (Figure 4f,g). However, we found here that combination of VER with MA exerted a cytotoxic effect in both cell lines. Compounds applied together were significantly more toxic than each of them used separately (Figure 4f,g). This indicates that pan-HSPA inhibition by VER can potentiate the cytotoxic effect of MA. Due to the physical properties of the JG-98 molecule (a fluorophore with a broad emission spectrum), we omitted a cytotoxicity analysis in JG98-treated cells.

### 3.5. HSF1 Contributes to Cytoprotection against MA Toxicity

HSF1 is a master activator of HSPs induction. We detected strong MA-induced phosphorylation of HSF1 at serine 326 in lung cancer cells (Figure 2b). A similar response of HSF1 to MA (Figure 5a) that was accompanied by overproduction of HSPA1 and HSPA6 expression and HSPA2 downregulation (Figure 5b) was also observed in the wild type (wt) MCF7 breast cancer cell line. The IC_50_ value of MA on MCF7 cells (5.109 µM; 95% CI, 4.73–5.487) was lower than that of NCI-H23 or NCI-H1299 cells (Figure 1b). Moreover, pan-HSPA inhibition potentiated the anticancer effect of MA in MCF7 cells (Figure 5c). Altogether, it seems that MA-induced HSF1-mediated cytoprotection can be a common mechanism that counteracts MA toxicity in various types of cancer cells. In order to examine to what extent HSF1-dependent transactivation pathways can contribute to MA resistance, we used HSF1-null MCF7 cells generated via the CRISPR/Cas9-mediated gene editing system (Figure 5d). We examined two HSF1-null gene-edited isogenic cell clones; as a control, we used a pool of isogenic cell clones that remained non-gene-edited after transfection with two gRNAs and Cas9 (CRISPR-CTL); we also used wt cells (Figure 5d). Of note, the growth rate of HSF1-null isogenic clones was lower than that of control cells (CRISPR-CTL and wt) under culture conditions used in our experiments (Appendix A).

It has been confirmed that the HSF1-null isogenic MCF7 clones used in this study are incapable of elevating the transcription of stress-inducible *HSPA1* and *HSPH1* genes after heat shock insult [31]. Here, we found that the removal of HSF1 has no influence on the basal levels of HSPA1 protein, but it was associated with elevated levels of HSPA2 compared to that of wt and CRISPR-CTL cells (Figure 5e). As expected, MA-induced activation of the HSR was totally blocked in HSF1-null cells, as no increase in the levels of stress-inducible HSPA1 and HSPA6 by MA were detected (Figure 5e). In addition, no reduction in HSPA2 levels was detected in HSF1-null cells after exposure to MA (Figure 5e). Thus, we found that removal of HSF1 completely prevented the MA–induced alterations in HSPAs expression.

Results of the alamarBlue^®^ and MTS assays indicate that the removal of HSF1 raised cell sensitivity to MA (Figure 5f,g). Surprisingly, even concentrations of MA that were non-toxic to control cells (2.5 µM) were sufficient to evoke a significant antiproliferative effect on HSF1-null cells. Thus, our results suggest that HSF1-directed HSR is necessary to provide a significant level of protection against MA cytotoxicity. Finally, we found that combination of MA with VER in HSF1-null MCF7 showed significantly higher antiproliferative effects than did single treatment (Figure 5g). This observation indicates that in MCF7 cells the basal levels of HSPAs are necessary to provide some residual level of protection against MA toxicity.

## 4. Discussion

In this work, we showed that MA activates the HSF1-orchestrated overproduction of the stress-inducible HSPA paralogs. Our results indicate that the basal intrinsic level of HSPAs, namely a joint activity of the stress-inducible and constitutively expressed paralogs, contribute to mechanisms of MA resistance. Our assumption is supported by results showing that pan-HSPA inhibition evoked superior anticancer efficacy compared to that of a selective blockade of a single HSPA paralog (including the main stress-inducible HSPA1). HSPAs inhibition sensitized lung and breast cancer cells to MA irrespective of their intrinsic resistance to MA since this effect was detected both in MA-sensitive and MA-resistant cells (Figure 4 and Figure 5). We also observed that the HSF1-mediated cytoprotective response rendered cells resistant to MA (Figure 5).

The global inhibition of HSPAs was found to reduce viability in various types of cancer cells [26,32,33,34]. Additionally, pan-HSPA inhibitors showed synergy with certain anticancer drugs. Their combination with bortezomib, a first-in-class proteasome inhibitor used as an oncological drug, showed synergistic effects on NSCLC cells in vitro [26] and in multiple myeloma in vitro and in vivo models [34,35]. Triple combination of HSPA inhibitor with cisplatin and 17-AGG, a HSPC inhibitor, showed a synergistic effect on bladder cancer cells [32]. However, HSPA inhibition was shown to decrease anticancer effects of platinum derivatives on NSCLC cells in vitro [26]. Therefore, our findings are important in the context of a potential future application of HSPA inhibitors in clinical settings. To date, there is no agreement in the field about whether a clinically relevant candidate for an HSPA inhibitor with anticancer activity should target a single selected HSPA paralog (the major stress-inducible HSPA1 preferentially) or instead target a whole group of highly homologous HSPA paralogs (both stress-inducible and non-stress-inducible). However, our results suggest that the second possibility may have great potential for future anticancer applications. It seems that pan-HSPA inhibition can target the redundant network of HSPA paralogs to overcome their joint pro-survival activity. In this context it is noteworthy that non-cancerous epithelial cells are relatively insensitive to pan-HSPAs inhibition [26,36].

Different molecular mechanisms can underlie anticancer effects of HSPAs inhibition. Disruption of vital pro-cancerous inter-protein interactions, examples of which are the HSPAs–BAG3 interaction [33] or HSPAs–DNAJ interaction [36], seems the most important. These mechanisms can trigger apoptosis in apoptosis-competent NSCLC cells, exert cytostatic or non-apoptotic death-promoting effects in apoptosis-defective NSCLC cells [26], or lead to perturbations in mitochondrial proteostasis, secretion of danger-associated molecular patterns (DAMP), and increased recruitment of immune cells in primary and metastatic models of colorectal cancer [36]. Given that HSPAs can interact with components of the immune system, a type of immunogenic HSPA inhibitor would potentially be useful to modulate anticancer immune responses.

In this work, we also confirmed that the HSF1-mediated HSR can effectively counteract the anticancer activity of MA. HSF1 is overexpressed or shows high nuclear accumulation in a broad range of tumors and tumor cell lines both in vitro and in vivo (review in [37]). It is generally assumed that HSF1 governs many aspects of cancer cell growth and metabolism [38,39]. It was shown that reduction in HSF1 levels and/or activity is suppressive to cancer cell growth [40,41,42,43]. Since inhibition of HSF1 appears to be a promising therapeutic strategy for cancer treatment, the first attempts to develop a targeted inhibitor of HSF1 activity were made using natural compounds with pleiotropic activity (e.g., quercetin or triptolide) [44]. Subsequently, several high-throughput screens for new HSF1 inhibitors have been performed, and some promising compounds were selected [45,46]. Since the selective targeting of HSF1 expression via genetic means results in a strong anticancer effect, developing a selective chemical HSF1 inhibitor may offer a promising future therapeutic option.

We found that MA-induced stress led to opposite changes in the levels of stress-inducible versus stress-non-inducible members of the HSPA family. Surprisingly, this dual response seems to be controlled by HSF1. The removal of HSF1 prevented not only an MA-induced increase in HSF1-dependent paralogs but also a drop in the levels of non-stress inducible ones (Figure 5). HSF1 knockout was also associated with the increase in the basal levels of HSPA2. Similar two-directional alterations in the protein levels of HSPAs were observed previously in NSCLC cells exposed to bortezomib [26]. Bortezomib induces polyubiquitinated protein accumulation that causes proteotoxic stress and activation of HSF1-mediated signaling. Thus, bortezomib and MA can generate analogous molecular responses, at least in terms of activation of the HSF1-mediated HSR. Although it seems paradoxical (HSF1 is a canonical activator of HSPs expression), this observation may suggest that HSF1 can act as a repressor of non-stress-inducible *HSPA* genes. Similar results showing that overproduction of HSF1 correlated with repression of the murine *HSPA2* gene expression were obtained in spermatogenic cells in transgenic mice [47]. As no direct in vivo interactions between HSF1 and the *HspA2* gene promoter was confirmed, it seems that HSF1 can affect *HSPA2* expression through some indirect mechanism [47].

It is well-documented that MA exhibits strong anticancer activity in vitro. MA was effective as a single agent against triple-negative breast cancer in nude mice [6]. MA was also shown to increase the efficacy of other anticancer strategies in animal models [48,49,50]. Dual combination of MA and paclitaxel was more potent than a single treatment against an anaplastic thyroid carcinoma model in vivo [51,52]. MA was also considered potentially useful for radiosensitization of a human pancreatic cancer model in vivo [53]. Thus, available data provide preclinical evidence for the potential use of MA in anticancer treatment. However, no clinical study has been performed with MA thus far. The reason being that MA’s ability to inhibit cancer cells growth may be partially dependent on pH conditions [54].

In this study we provided promising evidence that the anticancer effectiveness of MA is increased when combined with genetic ablation of HSF1 or chemical pan-HSPA inhibition. These observations may indicate that anticancer activity of natural compounds, such as MA, can be potentiated by inhibitors targeting cytoprotective mechanisms that limit their anticancer potential. Such a strategy could provide a promising therapeutic option, paving the way for further preclinical experiments.

## Figures and Tables

**Figure 1 cells-10-01418-f001:**
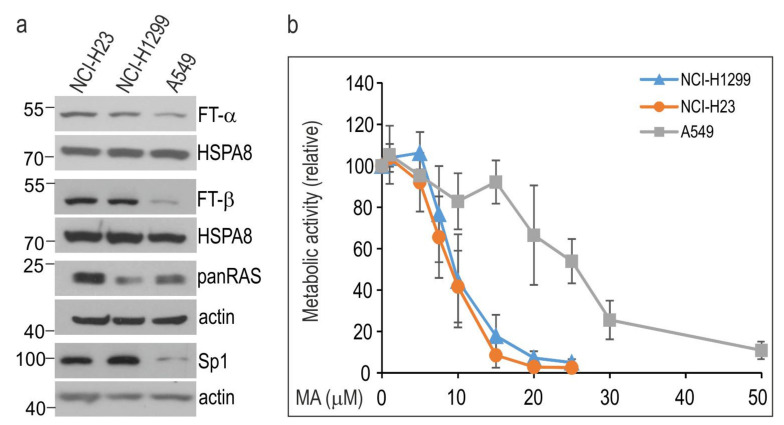
NSCLC cell lines show different sensitivity to manumycin A (MA). (**a**) The basal levels of MA targets in NSCLC cells. Representative immunoblots are shown (*n* ≥ 3); actin or HSPA8 were used as a protein loading control. Numbers on the left or right side of the blots indicate molecular weight of a protein size marker. (**b**) Dose-response curves of MA in NSCLC cell lines. Cell viability was measured following 72 h treatment with MA (0–50 µM) using an MTS assay. Results (mean ± SD from at least four independent measurements, each in three technical replicates) are expressed relatively to untreated control.

**Figure 2 cells-10-01418-f002:**
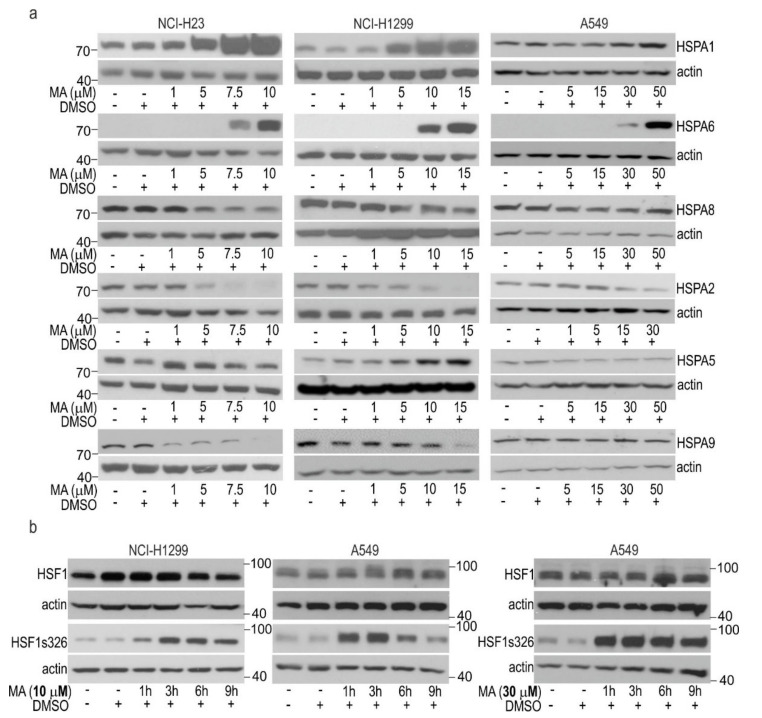
MA-induced changes in (**a**) the protein level of the HSPA paralogs, and (**b**) HSF1 phosphorylation in lung cancer cells. Levels of HSPA paralogs (**a**); total HSF1 (**b**); phosphorylated HSF1 (**b**) in cells non-treated or treated with MA for 24 hours (in (**a**)) or for 1–9 hours (h) (in (**b**)). Representative immunoblots are shown (*n* ≥ 3); actin was used as a protein loading control. Numbers on the left or right side of the immunoblot indicate the molecular weight of a protein size marker.

**Figure 3 cells-10-01418-f003:**
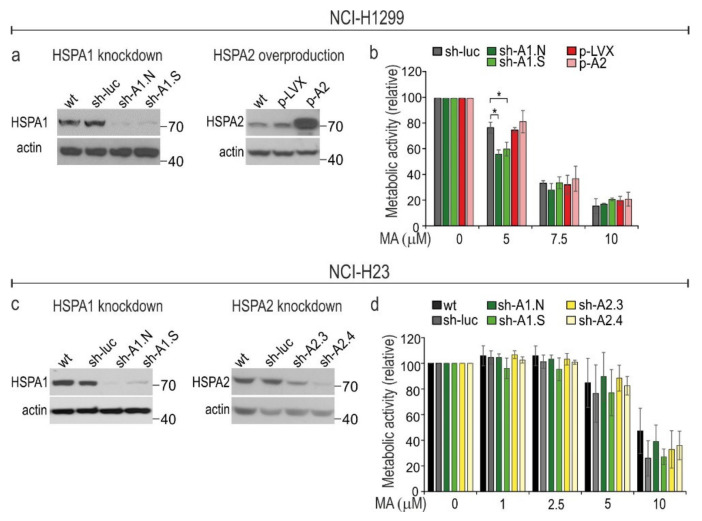
Manipulations in the protein levels of HSPA1 and HSPA2 have limited effect on sensitivity of NSCLC cells to manumycin A (MA). (**a**,**c**) Immunoblots showing levels of HSPA1 and HSPA2 in wild-type (wt) and lentivirally-modified cells; sh-luc control cells were transduced with a non-targeting shRNA-luc sequence; sh-A1.N and sh-A1.S cell lines were transduced with HSPA1-targeting shRNA sequences; control p-LVX cells were transduced with lentiviruses bearing the “empty” pLVX-Puro vector; p-A2 cells were transduced with pLVX-Puro plasmid encoding HSPA2 protein under the control of the CMV promoter; sh-A2.3 and sh-A2.4 cell lines were transduced with HSPA2-targeting shRNA sequences. Representative immunoblots are shown (*n* ≥ 3); actin was used as a protein loading control. These model cell lines were described in detail previously [26,27]. (**b**,**d**) Cell viability was measured using MTS assay after 72 h treatment with MA (0–10 µM). Results are expressed as mean ± SD in relation to untreated control (*n* = 4, each in three technical replicates, * *p* < 0.05, statistical significance was determined by two-tailed *t*-test). The numbers on the right side of immunoblots indicate molecular weight of the protein size marker.

**Figure 4 cells-10-01418-f004:**
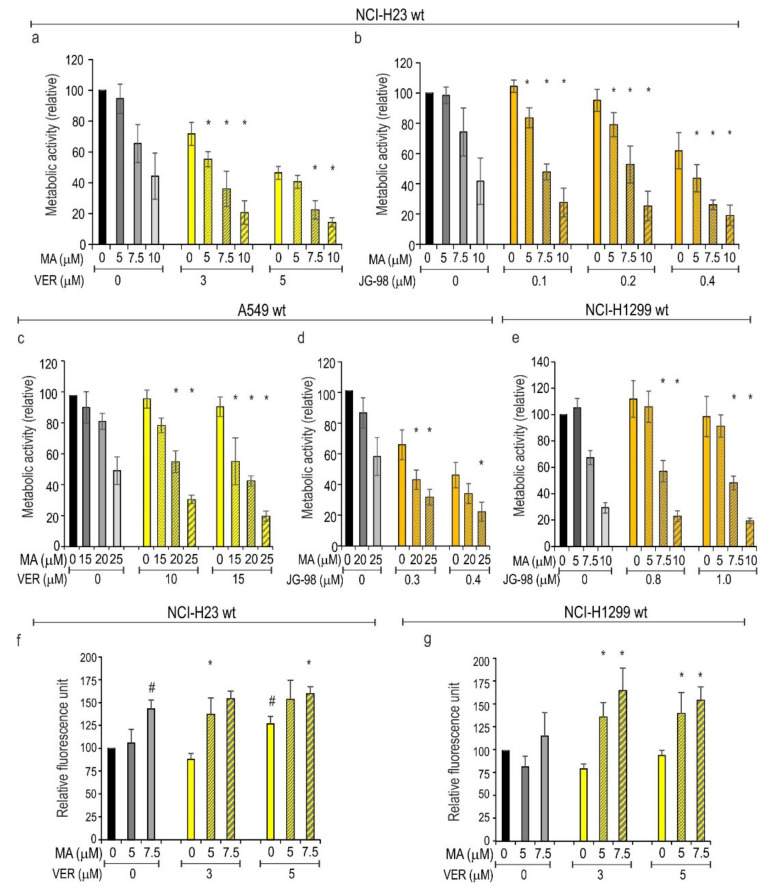
Effects of single or combined treatment (72 h) with (**a**,**c**,**f**,**g**) VER-155008 (VER) and manumycin A (MA) or (**b**,**d**,**e**) JG-98 and MA. Cell viability (**a**–**e**) was measured using MTS assay. Results are expressed in relation to the untreated control (mean ± SD, *n* ≥ 3, each in triplicate). Cytotoxicity (**f**,**g**) was evaluated using the CellTox™ cytotoxicity assay (mean ± SD, *n* ≥ 3, each in duplicate). Statistical significance was determined using two-tailed *t*-test; * *p* < 0.05, differences were considered significant if the value obtained for a double treatment was significantly different (*p* < 0.05) from the value obtained for each of the single treatments; # *p* < 0.05, statistical significance determined for single treatments (**f**,**g**).

**Figure 5 cells-10-01418-f005:**
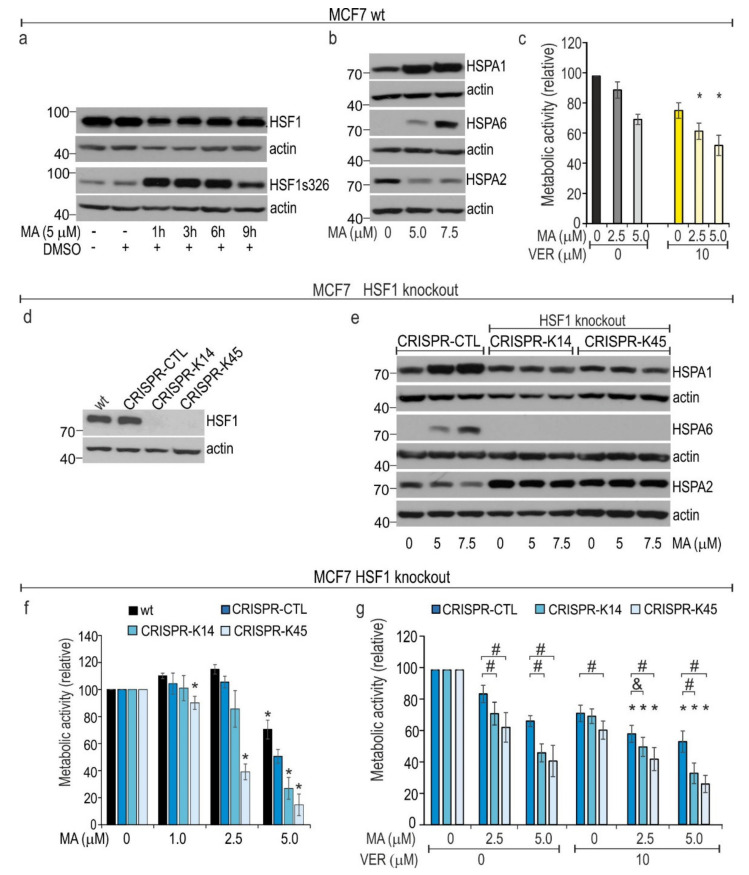
Effects of HSF1 knockout on response of MCF7 cells to manumycin A (MA). (**a**) Levels of total HSF1 and its phosphorylation at the serine 326 residue in cells exposed to MA for 0–9 h. (**b**) Levels of HSPA proteins in control and HSF1-null cells. (**c**) Viability of wild type (wt) cells after single and double treatment (72 h) with MA (0–5 µM) and/or VER-155008 (VER) (10 µM) assessed using an MTS assay. Results (mean ± SD; *n* = 5, each in two technical replicates) are expressed in relation to values obtained for non-treated cells. (**d**) Levels of HSF1 in wild-type (wt); control (CRISPR-CTL, a pool of non-edited isogenic clones); and two HSF1-null isogenic clones (CRISPR-K14, CRISPR-K45). (**e**) Levels of HSPA proteins in non-treated and MA-treated control and HSF1-null cells. (**f**) Viability of control and HSF1-null cells after MA (0–5 µM) treatment (72 h) assessed using the alamarBlue^®^ assay. Results (mean ± SD; *n* = 4, each in two technical replicates) are expressed in relation to values obtained for non-treated cells. (**g**) Viability of control and HSF1-null cells after single and double treatment (72 h) with MA (0–5 µM) and/or VER-155008 (VER) (10 µM) assessed using an MTS assay. Results (mean ± SD; *n* = 5, each in two technical replicates) are expressed in relation to values obtained for non-treated cells, control, and HSF1-null cells. In (**a**,**b**,**d**,**e**) representative immunoblots are shown (*n* = 3); actin was used as a protein loading control. Statistical significance was determined by two-tailed *t*-test. The numbers on the left side of blots indicate molecular weight of the protein size marker. In (**c**,**g**) * *p* < 0.05, differences were considered statistically significant if the value obtained for a double treatment was significantly different (*p* < 0.05) from the value obtained for each of the single treatments. In (**f**) * *p* < 0.05, statistically significant differences in relation to control CRISPR-CTL cells. In (**g**) # *p* < 0.05, statistically significant differences in relation to CRISPR-CTL cells, and & = 0.053.

## Data Availability

All materials and data are available upon reasonable request to the corresponding author.

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
