# Peer review of "Inhibition of the Heat Shock Protein A (HSPA) Family Potentiates the Anticancer Effects of Manumycin A"

_cells, 2021, doi:10.3390/cells10061418_

Round 1

Reviewer 1 Report

This manuscript by Sojka et al., reports the findings from a study testing the hypothesis that Heat Shock Proteins (HSPs) impede the anti-cancer effects of the natural antibiotic Manumycin A (MA) in vitro, a drug that is only used in preclinical settings so far. As hinted by the results of the proteomic study by Hu et al., Proteomics 2003 (ref#24), the authors demonstrate that the sensitivity of lung and breast cancer cells to MA can be potentiated by either inhibiting HSPA paralogs using pan-HSPA small molecule inhibitors (i.e. VER-155008 and JG-98) or by genetic depletion of the transcription factor HSF1, upstream activator of the heat shock response (HSR), respectively.

This work comes from a lab with long standing expertise in HSPs and cancer. The study is overall well conducted, and the results are clearly presented. I just have a few comments that I believe, if addressed, could reinforce the strength of this manuscript.

  • The rationale for this study remains elusive in the abstract. You could probably clarify why you thought HSPs could have such a counteractive role by adding a sentence after the first one and before “In this study...”.
  • The introduction is well laid out. The sentence lines 83-84 warning about why basically the hypothesis could be wrong, citing the authors’ paper #25 is a little vague. Could you please clarify? Maybe by adding “indirect” or “indirectly”?
  • All the viability assays are expressed in percentages from the untreated conditions, which is great to compare between conditions, however it masks the existing differences in the untreated conditions. Could the authors briefly comment on the cell densities of the untreated or show at least once that viability of the untreated cells was comparable between transfected cells of the same cell line?
  • In Figures 2b and 5a, can you please verify and show that total HSF1 protein is stable during the kinetics?
  • I am not sure to follow why the authors decided to overexpress HSPA2 in one cell line and knock it down in the other. Maybe adding a sentence to justify this would help.
  • The conclusions lines 268/269 stating that the toxicity of MA can be counteracted by a group of chaperones is slightly over-interpreted because the experiment was not about the overexpression or over-activation of HSPAs that would trigger resistance to MA. I would suggest to tone down the statement by rather mentioning that the toxicity of MA could be potentiated by inhibiting HSPAs.
  • It is a little surprising to see an unexpected switch from lung cancer cells to breast cancer cells for the last figure. Is there a meaningful reason as to why the CRISPR KO was not carried out in lung cancer cells? One could add a sentence in the beginning of paragraph 3.5 to justify this switch. Also, what is the IC50 of MA in MCF7? Please add it to the manuscript.
  • One experiment that would be nice to add to corroborate the last and most important finding of figure 5 would be to justify that the HSF1-mediated effects on MA sensitivity are dependent on transcription, for example by directly looking at mRNA levels and/or inhibiting transcription and doing RTqPCR.

Others:

  • Line 23: add "(HSF1)"
  • Line 26: HSR is not spelled out
  • Lines 105, 179, 180: in the non-round numbers, a period should be used for the decimals and not a comma.
  • Table 1 would be best located with the supplementary materials. Redundancy line 137.
  • Line 165: tailed
  • In the survival bar graphs, it is really hard to decipher the different conditions with the colored patterns. Plain colors would make it easier.
  • Line 245: NSCLC
  • Line 281: figure 5.
  • Line 347: mechanisms (plural)
  • Line 360: “cancer treatment, the first...”
  • Figure S1 legend: please explain that these densitometries correspond to figure 2a. Also, resolution of the text in the bar graphs appears very low.

Reviewer 2 Report

In this paper, Sojka et al. studied the capacity of mamumycin A (MA) to inhibit HSP expression and then to explain its potential anticancer properties.

While the aim of the paper is of interest, the experiments shown did not clearly prove the importance of HSPs/HSF1 in MA effects on cell viability.

The paper is divided into two parts:

  • in the first part, the authors studied the impact of MA on HSP and HSF1 expression and cell viability in three human lung cancer cell lines (Fig 1 to 3). They also studied the impact of shRNA targeting some HSPs or HSP inhibitors on MA-mediated loss of viability (Fig3 and 4).
  • in the second part, the authors studied the impact of MA on HSF1 and HSP expression and cell viability in a human mammary cancer cell line. They also studied the impact of CRISPR-mediated HSF1 depletion on MA-induced HSP expression and cell viability loss (Fig 5).

These two parts seem quite different and not so much linked: different cell lines, different targets (HSP or HSF1), different cell viability assays (MTS or alamarBlue).

Moreover, the effect of shRNA or CRISPR on cell viability was limited. It was only  observed with one dose of MA on 3 tested for shRNA. The two CRISPR against HSF1 did not show similar effect, one improving the other one having no effect on MA-mediated loss of viability (except for the highest dose). The effects observed when silencing HSP individually are not convincing while the effect of HSF1 silencing seemed to be promising. This is logical, as targeting HSF1 may result in targeting several HSPs. However, the discrepancies observed for CRISPR aganinst HSF1 should be solved, as here we cannot conclude about its role in MA effects.

Despite an effect on cell viability, the study would be strenghtened by studying other cancer cell features, such as cell death and/or cell cycle.

Round 2
